# An Overview of the Occurrence of Bioactive Peptides in Different Types of Cheeses

**DOI:** 10.3390/foods12234261

**Published:** 2023-11-24

**Authors:** Adriano Henrique do Nascimento Rangel, Débora América Frezza Villar de Araújo Bezerra, Danielle Cavalcanti Sales, Emmanuella de Oliveira Moura Araújo, Luis Medeiros de Lucena, Ana Lúcia Figueiredo Porto, Ítala Viviane Ubaldo Mesquita Véras, Ariane Ferreira Lacerda, Cláudio Vaz Di Mambro Ribeiro, Katya Anaya

**Affiliations:** 1Academic Unit Specialized in Agricultural, Federal University of Rio Grande do Norte (UFRN), Macaiba 59280000, Brazil; adrianohrangel@yahoo.com.br (A.H.d.N.R.); dvillar2@hotmail.com (D.A.F.V.d.A.B.); manu_moura9@yahoo.com.br (E.d.O.M.A.); luis_lucenaquim@yahoo.com.br (L.M.d.L.); 2Morfology and Animal Fisiology Departament, Rural Federal University of Pernambuco (UFRPE), Recife 55292901, Brazil; analuporto@yahoo.com.br; 3Federal Institute of Education, Science and Technology (IFRN), Currais Novos 59380000, Brazil; itala.mesquita@ifrn.edu.br (Í.V.U.M.V.); arianelacerda@yahoo.com.br (A.F.L.); 4School of Veterinary Medicine and Animal Science, Federal University of Bahia (UFBA), Salvador 40170110, Brazil; claudioribeiro@ufba.br; 5Faculty of Health Sciences of Trairi, Federal University of Rio Grande do Norte (UFRN), Santa Cruz 59200000, Brazil; katya.anaya@ufrn.br

**Keywords:** antioxidant, bioactivity, cheese, functional food, health, maturation, peptide

## Abstract

The search for improvements in quality of life has increasingly involved changes in the diet, especially the consumption of foods which, in addition to having good nutritional value, are characterized by offering health benefits. Among the molecules that trigger several beneficial responses are peptides, which are specific fragments of proteins known to produce positive effects on the human body. This review aimed to discuss the bioactive potential of peptides from cheeses. Studies show that the protein composition of some cheese varieties exhibits a potential for the release of bioactive peptides. The production of these peptides can be promoted by some technological procedures that affect the milk structure and constituents. The cheese maturation process stands out for producing bioactive peptides due to the action of enzymes produced by lactic acid bacteria. Thus, in addition to being proteins with high biological value due to their excellent amino acid profile, peptides from some types of cheeses are endowed with functional properties such as anti-hypertensive, antimicrobial, antioxidant, anticarcinogenic, opioid, and zinc-binding activities.

## 1. Introduction

The constant search for improved quality of life has been positively correlated with the consumption of foods with functional claims, which in addition to having good nutritional value, are also characterized by provision of health benefits. In this context, it is worth highlighting the action of bioactive peptides, which are specific fragments of proteins that can produce different positive effects on the body, whether biological, biochemical, and/or physiological [1].

Bioactive peptides consist of a sequence ranging from 5 to 30 amino acids, some of which are resistant to the digestive action of peptidases in the human gastrointestinal system, which is why they can be absorbed and pass into the bloodstream [2,3,4].

Peptides have numerous bioactivities that promote human health and prevent chronic diseases, producing benefits for the cardiovascular system (antithrombotic and anti-hypertensive effect), for the immune system (anti-inflammatory, antioxidant, and antimicrobial), the endocrine system (anti-obesity, appetite control, benefits associated with glucose, insulin, cholesterol, and triglycerides), and the nervous system [5,6].

Human nutritional needs are basically met through the consumption of meat and vegetable products. Within this diversity, milk and dairy products are widely consumed sources of protein and bioactive peptides, easily accessible and with low purchase value [6,7]. Studies indicate that cheese is a good source of bioactive peptides. Several studies have been conducted with a focus on the properties of dairy proteins, especially on the release of bioactive compounds during industrial processing or digestion [8,9,10,11,12]. All the benefits already discovered explain the incorporation of bioactive peptides derived from milk in formulating functional foods, dietary supplements, and pharmaceutical products that promote health, mitigate the risk of chronic diseases, and improve quality of life [5]. In this context, there are also products derived from milk such as cheese, which is a food widely consumed throughout the world, very diverse in terms of origin, raw material, production method, final composition, and benefits. Therefore, the following review aims to present few bioactivities of some varieties of cheeses due to the presence of peptides in their composition.

## 2. Cheeses as a Source of Bioactive Peptides

It is believed that cheese emerged in the fertile crescent between the Tigris and Euphrates rivers in Iraq 8000 years ago during the so-called agricultural revolution, which occurred with the domestication of plants and animals [13]. It originated from the classic principles of food preservation, such as lactic fermentation, i.e., reducing the amount of water and adding salt [14]. Cheese is a dairy concentrate consisting of proteins, lipids, carbohydrates, mineral salts, calcium, phosphorus, and vitamins, being one of the most nutritious foods known [14].

The raw material of cheese is milk, whose predominant protein composition is casein (representing 2.5 to 3.2% in bovine fluid milk), and approximately 80% of the total proteins in bovine milk [11]. Caseins are divided into four subgroups: alpha casein (48–50%), beta casein (33–36%), kappa casein (13–15%), and omega casein (2%) [11,15]. The remaining 20% comprises whey proteins, proteins from fat globules, growth factors, and enzymes [16].

Although no specific physiological property has been proposed for the entire milk casein system, several authors have already demonstrated that these proteins are a good source of peptides that can influence several beneficial physiological responses in the body, which are known as bioactive peptides [11,17,18,19]. According to their size, they are classified into small peptides (less than 7 amino acids, the most active, but challenging to analyze with conventional proteomic approaches), medium peptides (7–25 amino acids), and large peptides (more than 25 amino acids) [20].

These peptides are present in the primary structure of milk proteins. They remain inactive until their release by different mechanisms (Figure 1), such as through the proteolysis of caseins and whey proteins (β-lactoglobulin, α-lactalbumin, serum albumin, immunoglobulins, lactoferrin, and protease–peptone fractions) through physical and chemical treatments (thermal treatment, homogenization, application of pressure, and milk coagulation) [2,11,21]. They can also be released by gastrointestinal digestion, enzymatic hydrolysis by the action of plasmin and cathepsin D present in milk, and through microbial fermentation [21]. Naturally occurring lactic acid bacteria in milk are primarily responsible for the proteolysis that results in forming bioactive peptides [11].

Maturation is a complex process that varies from cheese to cheese, being very important for improving its characteristics since it modifies the chemical and physical properties of the cheese mass, influences the texture and consistency, and forms compounds that will be responsible for developing the characteristic flavor of each cheese variety. It consists of the final manufacturing stage and lasts weeks to more than two years, depending on the cheese produced. A series of biological, chemical, and biochemical factors occur in a complex succession during maturation under the action of lipolytic and proteolytic enzymes [14]. Several authors have already pointed out that the maturation process of cheeses can potentiate the formation and release of bioactive peptides, acting together with the enzymes produced by the initial and non-initial lactic acid bacteria [10,22,23,24,25,26].

This process encompasses numerous biochemical pathways involving proteolytic, lipolytic, and glycolytic processes. Many dairy cultures are highly proteolytic, leading to the accumulation of bioactive peptides in matured dairy products [27]. Depending on the type of dairy product, the level of naturally formed peptides in the matrix varies along with the balance between release and subsequent hydrolysis during maturation [28].

The presence of bioactive peptides has been demonstrated in Mozzarella cheeses aged for two weeks and in Parmigiano-Reggiano and Cheddar cheeses aged for more than two years [10,25,29]. A high presence of bioactive peptides was also observed in Canastra cheese [30] and Coalho [31,32], traditional Brazilian cheeses, and goatskin Tulum cheeses in Turkey [25].

There may be variations in the presence of peptides in the rind and inside the cheese. The study by Robinson et al. [33] demonstrated that there are substantial differences in proteolysis and consequently in the formation of peptides between the interior and the rind when analyzing Casu Marzu, Taleggio, Stilton Blue, and Mimolette cheeses. Differences in the microbial composition between manufacturing regions and the distinction in this microbiota inside and on the crust may explain the dissimilarity in peptide formation (and consequently in functional and organoleptic characteristics) in these two parts.

The physiological functions regulated by bioactive peptides vary according to the amino acid sequence and its location within the protein. These actions have been described by several studies that reproduced them in vitro using biochemical assays or in vivo with cells and animal/human models [34]. A compilation of the main functional properties investigated in bioactive peptides detected in some types of cheeses is illustrated in Figure 2 and described in Table 1.

## 3. Results for Biological Activities of Peptides Found in Cheeses

### 3.1. Anti-Hypertensive

Arterial hypertension (AH) is a non-transmissible chronic disease (NCD) defined by blood pressure levels, in which the benefits of treatment (non-medicated and/or medicinal) outweigh the risks. It is a multifactorial condition that depends on genetic, environmental, and social factors, characterized by persistent elevation of blood pressure (BP). Controlling arterial hypertension significantly reduces damage to target organs, such as the heart, brain, and kidneys, and medical and socioeconomic costs resulting from complications. The main anti-hypertensive drugs are adrenergic blockers, calcium channel blockers, diuretics, and drugs that intervene in the renin–angiotensin system [57].

The renin–angiotensin system is activated with the action of renin (a proteinase composed of approximately 350 amino acids, synthesized and stored in the juxtaglomerular cells of the kidney) on angiotensinogen (a glycopeptide with a molecular mass of approximately 60 kDa) in response to a drop in blood pressure, sodium depletion, or a reduction in plasmatic volume, releasing a peptide of 10 amino acids (angiotensin I) from its N-terminal portion [58]. Shortly after the formation of angiotensin I, two amino acids (His-Leu) are cleaved from its carboxyl end, forming the octapeptide angiotensin II, a potent vasoconstrictor. This reaction is catalyzed by the Angiotensin-Converting Enzyme (ACE) and almost entirely occurs in the lungs [59].

The anti-hypertensive effect of some cheese varieties occurs through the action of three peptides on the inhibition of ACE, thus producing a beneficial impact in individuals with high blood pressure [59]. The ACE inhibitor peptides are represented by different fragments and are called lactokinins when derived from whey proteins and casokinins when derived from casein. They commonly contain alanine, valine, and proline in their structure, suggesting that these amino acids play an essential role in this anti-hypertensive effect [60,61].

Okamoto et al. [62] were the first to report and describe ACE inhibition in cheese. Different ACE inhibition levels were observed in other in vitro studies on water-soluble peptide extracts [10,29,34,38,63,64,65] as well as in the ethanol-soluble fraction [66] of various types of cheeses.

ACE inhibitor peptides have been isolated from Italian cheeses such as Crescenza and Gorgonzola [35]. The inhibitory activity was also evidenced in other varieties of cheese such as Gouda, Blue, Edam, and Havarti in an experiment involving oral administration of peptides in rats with high systolic blood pressure, resulting in a decrease in blood pressure after 6 h after gastric intubation [38].

The relationship between the maturation process and the presence of anti-hypertensive peptides was demonstrated by Álvarez Ramos et al. [65], who reported maximum ACE inhibition in Gouda cheese aged for three months compared to the same cheese with a shorter aging period. Samples matured for more than three months showed reduced ACE activity, suggesting that the concentration of bioactive peptides increases with cheese maturation, but starts to decrease with the intensification of the proteolysis. The study conducted by Iwaniak et al. [37] demonstrated that Gouda matured for more than 60 days and composed of different β-casein levels showed intense ACE inhibitory action regardless of the β-casein content.

The ACE inhibitory activity in Manchego cheese produced from sheep milk and with different starter cultures fluctuated throughout the maturation period, decreasing in the first four months with a subsequent increase to maximum activity at eight months, and then a later decrease at 12 months of maturation of the cheese [36]. According to the authors, the active peptides are naturally formed in cheese depending on a delicate balance between their formation and their degradation by the proteolytic systems involved in cheese maturation.

Peptides with anti-hypertensive action were also detected in cheeses made with raw sheep milk produced in Southern Brazil, such as Roquefort, Feta, and Pecorino Toscano cheeses (matured for 60, 180, and 270 days), and in the Uruguayan cheeses, such as Pecorino Sardo (80, 120, and 160 days of maturation) and Cerrilano (aged for 90 and 120 days) [47].

The anti-hypertensive peptides Ile-Pro-Pro and Val-Pro-Pro were found in Swiss cheeses (Appenzeller, Tilsiter, Tête de Moine, Vacherin fribourgeois, Emmental, Gruyère, and Berner Hobelkäse) and increased during the ripening process, reaching 100 mg/kg after 4 to 7 months [67].

Helal and Tagliazucchi [26] studied six different cheeses (Karish, Feta-Type, Domoiati, RAS, Gouda, and Edam) obtained with different ingredients, thermal treatment, and maturation times. The authors identified anti-hypertensive peptides in all types of cheeses.

Although there are several studies demonstrating the functionality of peptides on the cardiovascular system, the vast majority of experiments analyzed the in vitro ability rather than the signaling action mechanisms, therefore, requiring much more research and multidisciplinary knowledge (biology, medicine, bioinformatics, and food technology) [34,68] to determine if and in what quantity the consumption of cheese could promote the desirable effects in the human body.

### 3.2. Antimicrobial

The discovery of novel antibacterial molecules is a crucial step in overcoming the great challenge posed by the emergence of antibiotic resistance [69,70]. More than 3100 antimicrobial peptides have been identified in various natural sources or predicted through the gene sequences that encode them [70,71,72]. These peptides vary in sequence and number of amino acids, generally from 3 to 50 amino acids in length, and molecular mass between 800 to 3500 Da [32,73,74].

Although the action mode of these peptides against microorganisms can occur through distinct mechanisms, it generally involves an association with the lipids of the plasmatic membrane of microorganisms, promoting increased permeability. Moreover, the fact that they have a positive electrical charge and are amphipathic enhances their solubility in an aqueous environment; thus, they are easily inserted into lipid membranes, triggering the death of target microorganisms [70,75,76].

Several antimicrobial peptides have been isolated from food sources, but most peptides with this potential are derived from milk proteins [11,70]. There are reports of antimicrobial fragments derived from αs1-casein [26], αs2-casein [26,74], β-casein [26,61], and κ casein [26,72]. Furthermore, there are peptides with antibacterial activity derived from α-lactalbumin and β-lactoglobulin [11].

Antimicrobial activity was found in two kinds of artisanal cheeses of Brazilian origin: Artisanal Coalho cheese and Canastra cheese. In a study conducted by Silva et al. [32] to evaluate the antimicrobial activity of artisanal Coalho cheese from the Agreste region of Pernambuco, in Brazil, it was found that the cheeses were also composed of peptides with molecular masses between 800 and 3500 Da, which is in the same range listed by the authors Papo and Shai [73] and López-Expósito et al. [74].

In a recent study in Brazil, we observed antimicrobial, antioxidant, and anti-hypertensive activities for bioactive peptides in artisanal Coalho cheese at different maturation times, and the maturation process enhanced these capacities, with a decline of observed activities at 60 days of maturation [31]. Dias et al. [41] identified peptides capable of inhibiting the action of microorganisms (*Bacillus subtilis* and *Enterococcus faecalis*) in artisanal Coalho cheese and reinforced their unique importance for cheese safety.

Canasta cheese matured for 9, 23, and 30 days showed fractions of peptides derived from αs1-casein and β-casein with antimicrobial activity against *Escherichia coli* bacteria at the minimum inhibitory concentration (MIC) of 15, 17, and 11 µg of proteins and soluble peptides/mL of solution, respectively [30]. Antimicrobial peptides were also isolated from Mozzarella, Italico, Crescenza, and Gorgonzola cheeses, with specific inhibitory action against the endopeptidases of *Pseudomonas fluorescens*, the microorganism responsible for compromising the technological and organoleptic characteristics of dairy products [35].

Basilicata et al. [7] found that buffalo milk mozzarella cheese released antimicrobial peptides derived from ĸ-CN (YYQQKPVA, f64-69; YYQQKPVA, f64-70) after gastrointestinal digestion, whose activity against *Escherichia coli* has already been mentioned in the literature. Peptides with antimicrobial bioactivity were also identified in Emmental cheese in a study carried out in vitro [42].

The antimicrobial activity in cheeses is an additional benefit during the production process since it enables a possible contamination reduction in dairy products, increasing the product’s shelf life. However, there are few studies on the occurrence or use of antimicrobial peptides in commercial dairy products. Apart from nisin, no other natural antimicrobial peptide is approved by the main regulatory agencies in the world as a food antimicrobial additive. Moreover, as most studies were carried out in vitro, studies in live models providing more information on action mechanisms regarding their therapeutic potential are needed [14].

### 3.3. Antioxidant

Oxidation processes are the leading cause of reduced shelf life of foods and raw materials in general and the main factor responsible for the aging of living organisms [24]. Oxidation is indispensable to aerobic life, being involved in energy production, phagocytosis, regulation of cell growth, intercellular signaling, and synthesis of important biological substances; therefore, free radical molecules that are naturally produced in tissues may protect the body from oxidative stress and damage [5,24].

Free radicals are neutralized by endogenous antioxidant mechanisms and also with the help of dietary antioxidants. Free radicals promote “oxidative stress” when in excess, engaging in unwanted reactions with DNA molecules, RNA, proteins, and other oxidizable substances. They promote damage that can contribute to aging and lead to degenerative diseases, such as cancer, atherosclerosis, and rheumatic arthritis (among others) [5,24]. The accumulation of free radicals can be caused by defects in mitochondrial respiration, arachidonic acid metabolism, activation–inhibition of enzymatic systems, or by exogenous factors such as pollution, smoking, alcohol intake, or even inadequate nutrition [77].

Several food peptides have been identified to possess antioxidant capacity, and their biological activity has been widely studied since the effect was first reported by Marcuse in 1960. Milk and its derivatives, which are essential foods for human development, have been also shown to be beneficial for defense against oxidation through several mechanisms [24,78].

The antioxidant capacity of peptides is related to the action mechanism of some amino acids present; thus, acidic and basic amino acids are considered metal ion chelating agents, and those that donate electrons and/or protons or that contain sulfhydryl groups present the ability to act as free radical scavengers [24].

Such peptides generally contain 5 to 30 amino acid residues, with tyrosine (Tyr), tryptophan (Trp), methionine (Met), lysine (Lys), cysteine (Cys), and histidine being examples of amino acids related to antioxidant activity [6,11]. Hernández-Ledesma et al. [79] found more significant antioxidant activity in peptides found in one of the fractions studied in fermented milk, and six of them had more than two proline residues. The high content of proline peptides could determine the antioxidant activity found in this fraction.

Tonolo et al. [80] demonstrated that milk-derived peptides, potentially capable of crossing the intestinal barrier, exert antioxidant activity through activation of the Keap1-Nrf2 signaling pathway. It promotes genes expression that encode the Sod1, Trx1, and TrxR1 antioxidant enzymes, increasing their respective production.

In addition, αs1- and αs2-casein and β-casein can be excellent sources of casein phosphopeptides (CPP), which are peptides that exhibit secondary antioxidant activity against the ferrous transition ion and scavenge and quench direct free radicals in aqueous and lipid emulsion systems [81,82].

Several studies have demonstrated the antioxidant activity of peptides from some cheese varieties, such as cheddar cheese [43,44], artisanal Coalho cheese [31,32], queso fresco cheese [45], Parmigiano-Reggiano cheese [10], cottage cheese [46], Karish, Feta-type, Domiati, Ras, Gouda, and Edam cheeses [26], and feta, pecorino toscano, roquefort, pecorino sardo, and cerrilhano cheeses made with raw sheep milk and matured at different times [47]. Furthermore, the release of antioxidant peptides from the action of pepsin in separate whey and casein fractions from goat’s milk was identified [83].

Gupta et al. [43] found the antioxidant property in cheddar cheese with increased bioactivity when matured for four months, but with a subsequent reduction in the 5th month. According to the author, the increase in peptide activity during maturation is due to the action of proteolytic enzymes from adjunct cultures, and the decrease in activity indicates that these peptides were not resistant to additional proteolysis.

Silva et al. [32] found the presence of 67 peptides with molecular masses ranging from 800 to 3500 Da in studying the bioactive properties of water-soluble peptides extracted from artisanal Coalho cheese made in different municipalities in the Agreste region of Pernambuco, Brazil. In addition, they verified that all extracts of peptides soluble in water at a concentration of 17.5 mg of peptides/mL showed antioxidant activity, with effects varying from 66.27% to 91.1%, depending on the cheese production site [32].

Antioxidant capacity was also demonstrated in whey of the Mexican white cheese known as panela, but at a maximum 1,1-Diphenyl 2-picrylhydrazyl (DPPH) sequestering capacity of only 26% [48].

The digestive process promotes activating peptides that were inactive in the original protein molecule and can modulate physiological functions when released, either through interaction with specific receptors or by inducing physiological responses [84]. However, peptides present in food are subject to further hydrolysis when exposed to digestive processes and may lose some functional characteristics [10]. When studying the biological and in vitro digestion potential of soluble peptides obtained from various kinds of cheese, Santos et al. [85] found that bovine and buffalo Coalho cheese, gorgonzola, mozzarella, cheddar, and ricotta had antioxidant activity both before and after in vitro digestion.

The antioxidant activity can be altered according to the type of rennet used in cheese manufacturing. This was well demonstrated by Timón et al. [86], who identified antioxidant peptides in hard cheeses made from cow’s milk with different types of rennet (vegetable, animal, and microbial) and found that cheese made from animal rennet had lower antioxidant activity in a study performed in vitro when compared to microbial rennet.

### 3.4. Immunomodulatory Effects

Some bioactive peptides have the ability to strengthen the human immunological system. The modulating function of these peptides ranges from increasing the response of immune cells (i.e., T and B lymphocytes) against invaders, to regulating the production of inflammatory cytokines linked with autoimmune diseases and inflammation.

Thus, when ingested and absorbed by the body, immunomodulatory peptides can help maintain the balance of the immune system in several ways, depending on the individual’s health conditions, habits, and diet.

Another relevant action performed by peptides derived from cheese is the immunomodulatory activity. According to Clare et al. [87], peptides derived from ß-casein stimulate phagocytosis by macrophages.

The C-terminal portion of peptides derived from ß-casein increases the proliferation of mouse lymphocytes in vitro experiments, with fragments 192 to 209 responsible for modulating immune functions [88]. This property was observed for molecules formed during the maturation of Comté and Grana Padano cooked cheeses [49,50,51]. Peptides with immunomodulatory activity have also been found in other types of cheese: Parmigiano-Reggiano [3,10], Chinese Rushan and Naizha [40], Goatskin Tulum cheese [25], Edam, Gouda, Karish, and Ras [26].

### 3.5. Zinc Binding

Another verified bioactivity is the ability to bind with zinc. Phosphorylated peptides from α1 and α2 caseins and ß-casein can form soluble complexes with minerals such as calcium, iron, and zinc in the intestinal pH, modulating their bioavailability. These peptides that act as mineral carriers are known as casein phosphopeptides (CPPs) and can be released “in vitro” or “in vivo” by enzymatic digestion of dairy products or during their processing [89,90].

Silva et al. [32] found that the peptides from artisanal Coalho cheese showed considerable zinc-binding activity, so the consumption of this cheese could increase the availability of this mineral to the body. Zinc plays a key role in the function of several enzymes and participates in cell division, gene expression, physiological processes, such as cell growth and development, and gene transcription [91].

### 3.6. Other Beneficial Effects

In the study conducted by Yasuda et al. [52], six of the twelve commercial cheese extracts tested (Montagnard, Pont-l’Eveque, Brie, Camembert, Danablue, and Blue) showed strong growth inhibition activity and induced DNA fragmentation in HL-60 cells. Long-term matured blue cheese was able to cause significant suppression of cell growth and induction of apoptosis, DNA damage, and morphological alteration of the nucleus in HL-60 cells. These results suggest the potential of long-term ripening cheeses to develop antiproliferative activities. Indeed, aqueous extracts of peptides from cheddar cheese made from buffalo and cow milk were also evaluated for antiproliferative activity in different cancer cell lines (colon and lung cancer) [53], and it was verified that the extracts collected in the 4th and 5th months of maturation reduced cell viability in a dose-dependent manner.

The literature also reports the opioid property of milk-derived peptides through binding to specific target cell receptors in an agonistic or antagonistic manner [34,92]. ß-casomorphins from ß-casein are the most studied opioid receptor ligands [59]. Brie [54], Gouda [26,38], gorgonzola, goat cheese, taleggio, fontina, cheddar, and Grana Padano [55,56], Parmegiano-Reggiano [39], and Crescenza [35] are among the cheeses which present β casomorphins.

In addition, the results from a study by Sánchez-Rivera et al. [92] revealed that “the in vivo effect on systolic blood pressure of the studied αs1-casein peptides is mediated by interaction with opioid receptors and the anti-hypertensive activity of casein hydrolysate can be very likely ascribed to them with the possible contribution of other mechanisms”.

It is important to emphasize that in order to exert physiological action, the peptides must be absorbed intact through the intestinal barrier and reach the bloodstream, overcoming hydrolysis by brush border peptidases during transepithelial transport [2]. The intestinal hydrolysis of food-derived peptides has been demonstrated in the Caco-2 cell monolayer model [89,93], and although their fragments can be found inside the intestinal cells, it is uncertain if they still maintain the original biological activity or display novel activities after hydrolysis [90]. Therefore, it is paramount that the molecular mechanisms by which bioactive peptides cross the intestinal epithelium and exert bioactivities on target tissues are investigated at the cellular and molecular levels [94].

Helal et al. [26] identified a multifunction fragment YPFPGPI from α-casein (60–66) in Gouda cheese with the following activities: anti-diabetic (DPP-IV-inhibitor), antioxidant, opioid, immunomodulator, anti-cancer, and anxiolytic. Among the five types of cheeses, only feta cheese displayed the fragment HIRL from ß-lactoglobulin (146–149), which solely presented anxiolytic activity.

Moreover, several multifunction bioactive peptides were also identified in Parmigiano-Regiano cheese matured for 12 months. Only one of them showed anxiolytic function, corresponding to the YLGYLEQLLR fragment from α-S1-casein (91–101 fraction), and one showed caseinophospheptide activity attributed to the RELEELNVPGEIVESLSSSEESITR fragment from ß-casein (1–25) [3].

## 4. Future Perspectives on Consumers’ Preference for Functional Foods

The health and well-being of human beings are highly influenced by their eating habits. The recent discoveries and the easy access to food information have led to the growth of demand for foods with functional properties.

The nutritional properties of milk and its derivatives are currently no longer the only criterion of choice at the time of buying such products. The preference and purchase intention of consumers for enriched milk and dairy products, with prominent functional properties for the immune system and other health declarations has increased in recent decades. This tendency is reflected in the current diversity of products available on supermarket shelves to meet this demand.

Understanding this new view of consumers about functional foods, including cheese and other dairy products, will allow food producers to be more aligned with the expectations of the consumer market.

Given the context of recent discoveries about the functional properties found in dairy and the massive broadcast of this information to society, there has been a change in people’s perception and preference for dairy foods with proven functional factors, instead of only caring about the nutrition facts. Thus, in the coming years, consumers preference may be more directly linked to the following: I—improvement in immunity; II—reduction in gastrointestinal discomfort and allergic reactions; III—prebiotic and probiotic action; IV—disease prevention (i.e., coronary and intestinal diseases, cancer, etc.); V—guarantee of longevity and quality of life; VI—nutrient absorption efficiency (i.e., in high-performance athletes and the older adult population), and VII—ease of product access and practicality in everyday life.

Thus, we the scientific community, primary producers, beneficiaries, and retailers of dairy products as a whole must pay greater attention to this change in consumers’ preference standards.

## 5. Final Considerations

Cheeses constitute an excellent matrix for the release of biologically active peptides, especially those made with raw milk and subjected to maturation. Lactic acid bacteria starters produce a range of peptide fragments derived from milk proteins that have anti-hypertensive, antimicrobial, antioxidant, anticarcinogenic, opioid, anxiolytic, and zinc-binding activities. However, it remains a challenge to determine the minimum amounts of regular cheese consumption that would result in positive effects on the human body.

In-depth studies to elucidate the mechanisms of action of the absorption of bioactive peptides by the digestive system will favor generating valuable knowledge about possible health benefits related to cheese consumption. From an economic point of view, the development of research in this area also adds value to cheeses, aiming to meet consumers’ desires for healthy foods with beneficial properties.

## Figures and Tables

**Figure 1 foods-12-04261-f001:**
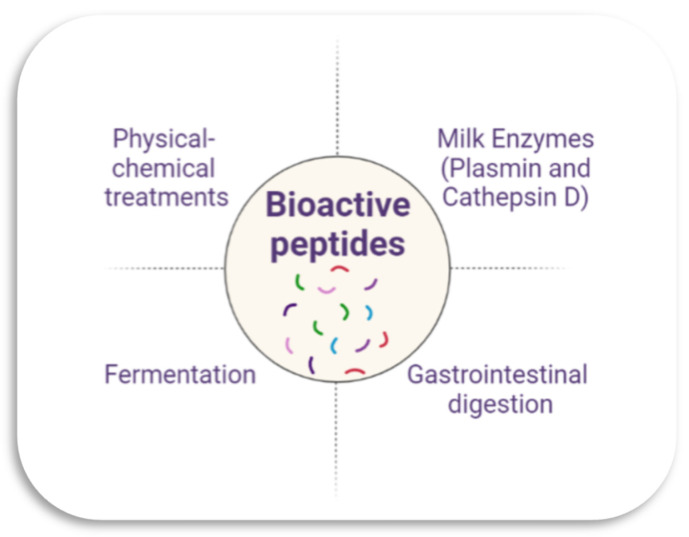
Main formation mechanisms of bioactive peptides from caseins.

**Figure 2 foods-12-04261-f002:**
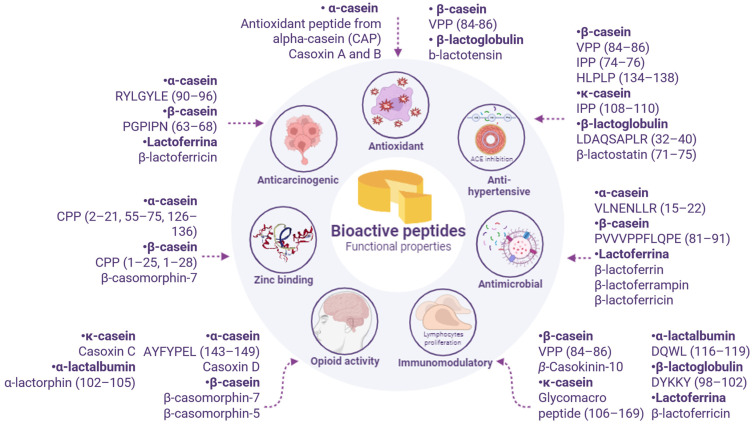
Illustration of results. Protein fraction of original and main functional properties described for bioactive peptides detected in some types of cheeses and is based on authors cited in Table 1 and Egger and Ménard [5].

**Table 1 foods-12-04261-t001:** Main functional properties described for bioactive peptides detected in some types of cheeses.

Functional Property	Cheese Type	References
Anti-hypertensive	Crescenza and Gorgonzola	[35]
	Artisanal Coalho cheese	[31]
	Gouda	[36,37]
	Blue, Edam, and Havarti	[38]
	Parmigiano-Reggiano	[3,10,39]
	Chinese Rushan and Naizha	[40]
	Goatskin Tulum cheese	[25]
	Karish, Feta-type, Domiati, Ras, Gouda, and Edam	[26]
Antimicrobial	Artisanal Coalho cheese	[31,32,41]
	Emmental	[42]
	Canastra	[30]
	Parmigiano-Reggiano	[3,10]
	Chinese Rushan and Naizha	[40]
	Buffalo mozzarella	[7]
	Goatskin Tulum cheese	[25]
	Karish, Feta-type, Domiati, Ras, Gouda, and Edam	[26]
Antioxidant	Cheddar	[43,44]
	Fresco	[45]
	Parmigiano-Reggiano	[3,10]
	Cottage cheese, Mozzarella, Ricotta, and Gorgonzola	[46]
	Feta, Pecorino Toscano, Roquefort, Pecorino Sardo, and Cerrilhano	[47]
	Artisanal Coalho cheese	[31,32]
	Mexican white cheese whey	[48]
	Chinese Rushan and Naizha	[40]
	Goatskin Tulum cheese	[25]
	Karish, Feta-type, Domiati, Ras, Gouda, and Edam	[26]
Immunomodulatory	Comté and Grana Padano	[49,50,51]
	Parmigiano-Reggiano	[3,10]
	Chinese Rushan and Naizha	[40]
	Goatskin Tulum cheese	[25]
	Edam, Gouda, Karish, and Ras	[26]
Anticarcinogenic	Montagnard, Pont-l’Eveque, Brie, Camembert, Danablue, and Blue	[52]
	Cow and buffalo milk cheddar	[1,53]
	Parmigiano-Reggiano	[3]
	Goatskin Tulum cheese	[25]
	Domiati, Edam, and Gouda	[26]
Opioid	Parmigiano Reggiano	[3]
	Domiati, Edam, Feta-type, and Ras	[26]
	Gouda	[26,38]
	Crescenza	[35]
	Brie	[54]
	Gorgonzola, Caprino, Taleggio, Fontina, Cheddar, and Grana Padano	[55,56]
Zinc binding	Artisanal Coalho cheese	[32]

## Data Availability

Data will be made available on request.

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
