# Peer review of "An Overview of the Occurrence of Bioactive Peptides in Different Types of Cheeses"

_foods, 2023, doi:10.3390/foods12234261_

Round 1
Reviewer 1 Report
Dear Authors
The manuscript (Ms ID: MDPI_Foods_-2551034 as the review entitled “An overview of the occurrence of bioactive peptides in different types of cheese” written by Adriano Henrique et al. is summarized the bioactive peptides focusing on antihypertensive, antimicrobial, anticarcinogenic, antioxidant, immunoregulatory, opioid, and zinc-binding derived from various type of milk and cheese.
Although the manuscript provides a comprehensive summary of the dominant actions reported on the bioactive peptides derived from distinct types of cheese and their materials of milk, it is totally needed reconsideration to construct a more structured. It seems to be just a series of short sentences citing references. It is recommended that the manuscript should be reconsidered the structure. In this manuscript, I wonder why there are so many co-authors on this manuscript preparation. Then, I confirmed the Author contributions that is the end of the manuscript. Why do most authors contribute methodology, data curation, formal analysis, and investigations? This manuscript is a review but not an article. There is no mention of data curation and discussion of methodology cited references in this manuscript. It is recommended, therefore, that the manuscript is required major revision for publication in this journal as the review article.
Minor comments:
1. Keywords
The keywords should be reconsidered because there is not contain “cheese” and “peptide” which are important words in this manuscript.
2. Introduction
The described introduction is very short and it seems to be insufficient to introduce the outline of this review.
3. line 113: Please correct “Figure 02” and “Table 01” to “Figure 2” and “Table 1”.
4. Table 1: Please correct “Anti-hipertensive” to “Anti-hypertensive”.
5. line 139: Please correct “Angiotensin-Converting Enzyme (ACE)” to “ACE” because it appeared 2nd time.
6. line 120: Please unify usage to either “anti-hypertensive” or “antihypertensive”.
7. line 193 and 206: The molecular masses of bioactive peptides are reported between 800 to 3,500 Da by citing references #55 and #56 in line 193. However, in line 206, it is cited reference #31. Hence, these synonymous sentences have different citations. Please explain or correct it.
8. The name of microorganisms should be written in Italics throughout the manuscript.
9. Almost no specific peptide examples are described throughout the manuscript.
10. line 378-384: The statement in this paragraph seems to be inappropriate for this review because the relationship between the bioactivity of peptides and Covid-19 has never been investigated.
11. There are few statements regarding “zinc-binding” and “immunomodulate” throughout this manuscript.
12. line 412-417: Author Contributions: Where is the methodology, data curation, and formal analysis performed and described in this manuscript?
13. For the references, please unify the format of references.
Author Response
September 25, 2023.
Foods
Dear Reviewer
Thank you very much for taking the time to review this manuscript.
Their comments were fully appreciated. The manuscript was revised and a broader reformulation of the text was carried out. We also did another round of editing in professional languages.
All edits suggested and other revisions are marked in the Author's Reply to the Review Report (Reviewer) file. Please see the attachment.
Best regards.
Corresponding author

Reviewer 2 Report
The article does not cover all the recent publications in the area of bioactive peptides in cheeses, for example, anti-diabetic peptides in cheeses and some others.
Author Response

(The authors gave the same response as above.)

Reviewer 3 Report
The manuscript is a review on the presence of bioactive peptides in cheeses.
The authors must make an analysis of the information. So far the manuscript contains a collection of article abstracts. For example, in the section where they review antioxidant activity, several works are mentioned, but they do not follow a logical sequence. The same happens in the other activities section.
In addition to this, the authors must review the entire manuscript, there are several errors in the way of citing.
Finally, they should check the references section to match the journal format. Make sure that the scientific names are written in italics.
Line 102 change into Author [32]
Line 145 the reference must be as number
Reference 89. Lactoccus must be in italics
The language is comprehensive, however, the manuscript must be reviewed for grammar.
Author Response

(The authors gave the same response as above.)

Reviewer 4 Report
In this paper, the authors described the state of the art of biopeptides in cheese. The review is interesting because it gives an overview of the subject and emphasizes the presence and activity of peptides.
In my opinion, the manuscript is well written and clear, but it has errors typographic, that is necessary minor corrections are required. I sent the revised document indicating the corrections

Author Response

(The authors gave the same response as above.)

Round 2
Reviewer 1 Report
Dear Authors
The responses to the first comment of the review were confirmed to be improved in this paper. Leave the final decision to the editor of this journal.
Reviewer 2 Report
The article could be published in its format.